# Efficient Charge Separation and Transport in Fullerene-CuPcOC_8_ Donor–Acceptor Nanorod Enhancing Photocatalytic Hydrogen Generation [note 1]

**DOI:** 10.3390/nano14030256

**Published:** 2024-01-24

**Authors:** Zihui Hua, Bo Wu, Yuhe Zhang, Chong Wang, Tianyang Dong, Yupeng Song, Ying Jiang, Chunru Wang

**Affiliations:** 1Beijing National Laboratory for Molecular Sciences, Key Laboratory of Molecular Nanostructure and Nanotechnology, Institute of Chemistry, Chinese Academy of Sciences, Beijing 100190, China; zhhuaa@iccas.ac.cn (Z.H.); fengyu971008@163.com (Y.Z.); wangchong@iccas.ac.cn (C.W.); dongtianyang006100@163.com (T.D.); jiangying@iccas.ac.cn (Y.J.); 2University of Chinese Academy of Sciences, Beijing 100049, China; songyupeng21@iccas.ac.cn; 3Key Laboratory of Photochemical Conversion and Optoelectronic Materials and CityU-CAS Joint Laboratory of Functional Materials and Devices, Technical Institute of Physics and Chemistry, Chinese Academy of Sciences, Beijing 100190, China

**Keywords:** fullerenes, internal electric field, charge separation, donor–acceptor structure, photocatalytic hydrogen generation

## Abstract

Photocatalytic hydrogen generation via water decomposition is a promising avenue in the pursuit of large-scale, cost-effective renewable hydrogen energy generation. However, the design of an efficient photocatalyst plays a crucial role in achieving high yields in hydrogen generation. Herein, we have engineered a fullerene-2,3,9,10,16,17,23,24-octa(octyloxy)copper phthalocyanine (C_60_-CuPcOC_8_) photocatalyst, achieving both efficient hydrogen generation and high stability. The significant donor–acceptor (D–A) interactions facilitate the efficient electron transfer from CuPcOC_8_ to C_60_. The rate of photocatalytic hydrogen generation for C_60_-CuPcOC_8_ is 8.32 mmol·g^−1^·h^−1^, which is two orders of magnitude higher than the individual C_60_ and CuPcOC_8_. The remarkable increase in hydrogen generation activity can be attributed to the development of a robust internal electric field within the C_60_-CuPcOC_8_ assembly. It is 16.68 times higher than that of the pure CuPcOC_8_. The strong internal electric field facilitates the rapid separation within 0.6 ps, enabling photogenerated charge transfer efficiently. Notably, the hydrogen generation efficiency of C_60_-CuPcOC_8_ remains above 95%, even after 10 h, showing its exceptional photocatalytic stability. This study provides critical insight into advancing the field of photocatalysis.

## 1. Introduction

The photocatalytic decomposition of water, a pivotal process enabling hydrogen generation from solar energy, stands as a promising solution to address the global energy crisis [1,2,3,4]. Since the pioneering work on utilizing TiO_2_ for hydrogen generation in 1972 [5], the primary research focus in this field has centered on inorganic semiconductors [6,7,8]. There are inherent defects in most inorganic semiconductor photocatalysts, such as wide forbidden bandwidths, rapid photogenerated carrier recombination, inefficient utilization of sunlight, and photo-corrosion during prolonged light exposure [9]. These restrict their further development and application. Conversely, there has been a recent surge of interest in organic semiconductor photocatalysts owing to their diverse functionalities, various synthesis methods, tunable energy bands, and wide absorption range [10,11,12]. Various organic semiconductor photocatalysts, including metal–organic frameworks (MOFs) [13], covalent organic frameworks (COFs) [14], graphitic carbon nitride (g-C_3_N_4_) [15], and linear conjugated polymers [16], have been utilized in the photocatalytic decomposition of water to produce hydrogen. The main challenge lies in the rational design of highly efficient photocatalysts. It includes the method to optimize their light trapping capability, efficient charge separation and transport, and long-term stability [17].

The donor–acceptor (D–A) structures in organic photocatalysts provide an effective strategy to enhance their catalytic activity [18,19]. Induced by the distinct electron affinities of the donor and acceptor units, the electron push–pull effect facilitates the directed migration of photogenerated electrons from the donor to the acceptor unit. It promotes the separation of photogenerated electrons and holes. This narrowing of the energy band gap of organic photocatalysts through the recombination of the highest occupied molecular orbitals (HOMOs) and the lowest unoccupied molecular orbitals (LUMOs) effectively enhances their light absorption capacity [20]. Moreover, the uneven distribution of positive and negative molecular charges leads to the formation of significant molecular dipole moments [21]. Notably, the magnitude of the molecular dipole moment exhibits a positive correlation with the strength of the internal electric field (IEF). The deliberate establishment of a robust IEF within a molecule effectively facilitates the separation and transportation of photogenerated carriers, making it one of the most efficient methods to enhance photocatalytic activity [22,23,24].

Fullerenes possess a three-dimensional symmetric structure with exceptional electron-accepting capabilities and commendable physical and chemical stability [25,26]. Copper phthalocyanine (CuPc) is an excellent organic pigment with high chemical stability to light and temperature [27]. CuPc is an effective donor material with remarkable tunability and high visible light absorption [28,29]. Due to these exceptional properties, a series of CuPc photocatalysts have been employed in the field of photocatalytic hydrogen generation [28,30,31,32]. However, the current efficiency of most Pc-based photocatalysts for hydrogen generation is unsatisfactory. In this study, a novel donor–acceptor (D–A)-type photocatalyst C_60_-CuPcOC_8_ was devised with the aim of enhancing photocatalytic hydrogen generation. C_60_-CuPcOC_8_ demonstrates exceptional hydrogen evolution activity, achieving a rate of 8.32 mmol·g^−1^·h^−1^. The introduction of octoctyloxy modification to CuPc is aimed at enhancing the solubility of phthalocyanine in organic solvents and allowing for the fine-tuning of the physicochemical properties of phthalocyanine derivatives [33]. The non-covalently supramolecular photocatalyst C_60_-CuPcOC_8_ was obtained by using the liquid–liquid interface precipitation method. The hydrogen generation working principal diagram of the C_60_-CuPcOC_8_ assembly is illustrated in Figure 1. C_60_-CuPcOC_8_ assembly exhibits a broad spectral response range spanning from 300 to 800 nm, allowing efficient utilization of solar energy. Additionally, the C_60_-CuPcOC_8_ manifests a robust IEF owing to the D–A interaction. This robust IEF promotes the efficient separation and transport of photogenerated charges. It can then be expected that C_60_-CuPcOC_8_ photocatalytic hydrogen generation activity will be significantly enhanced. 

## 2. Experimental

### 2.1. C_60_-CuPcOC_8_ Fabrication

The C_60_-CuPcOC_8_ assembly was prepared via the liquid–liquid interfacial precipitation method as shown in Figure 2. Initially, C_60_ and CuPcOC_8_ powders were dissolved in homo-trimethylbenzene. Subsequently, the solutions were sonicated for 30 min at room temperature. The sonicated solution was then mixed and filtered. Following this, 15 mL isopropanol was added to 5 mL C_60_-CuPcOC_8_ solution. The mixed solution was left to stand for three hours to obtain a precipitate. The resulting precipitate was then filtered and washed three times with methanol. Finally, the desired C_60_-CuPcOC_8_ assembly product was obtained after the vacuum drying. The self-assemblies of C_60_ and CuPcOC_8_ were achieved using an identical method.

### 2.2. C_60_-CuPcOC_8_ Characterizations

The powders for assembly or self-assembly were evenly distributed in a sample well. The phase structure was determined using a powder X-ray diffractometer (PXRD) (D/max2550VB/PC, Rigaku, Tokyo, Japan) equipped with Cu Kα radiation. The morphological and structure analysis was conducted by field emission scanning electron microscopy (FE-SEM) (Regulus 8100, Hitachi, Tokyo, Japan), transmission electron microscopy (TEM) (HT-7700, Hitachi, Tokyo, Japan), and high-resolution transmission electron microscopy (HRTEM) (JEM-2100F, JEOL, Tokyo, Japan). To conduct SEM testing, the prepared assembly/self-assembly was dispersed in ethanol to form a suspension and then added dropwise onto a silicon wafer. Similarly, TEM testing was performed by adding the suspension dropwise onto a copper microgrid. The UV-Vis diffuse reflectance spectra (DRS) were acquired with a Lambda 1050 UV/VIS/NIR Spectrometer (PerkinElmer, Waltham, MA, USA). The tests were performed with an integrating sphere attachment. The integrating sphere attachment comprised a spherical cavity with an inner wall coated with a highly reflective substance, and BaSO_4_ was employed as the standard reflectance material.

### 2.3. Photocatalytic Hydrogen Generation

The photocatalytic hydrogen generation was conducted employing a 300 W xenon lamp (CEL-HXUV300, CeAulight, Beijing, China) in conjunction with an all-glass automated online trace gas analysis system (Perfect Light Labsolar-6A, PerfectLight, Beijing, China). The dispersion of 5 mg photocatalyst powder in 100 mL of de-ionized water was achieved through ultrasonic treatment. 

The ascorbic acid (AA), at a concentration of 0.2 mol·L^−1^, was employed as a hole sacrificial agent. The hole sacrificial agent can consume photogenerated vacancies. Additionally, 3 wt% Pt was loaded as a co-catalyst on the photocatalysts by photodeposition. Photodeposition is a technique based on light-induced electrochemistry. The process of photodeposition of Pt depends mainly on the photoelectrochemical reaction induced by photocatalysts. The equation for the reductive photodeposition of Pt is as follows.
(1)$$PtCl_{6}^{2-}\left(aq\right)+4e^{-}\to Pt^{0}(s)+6Cl^{-}$$

Evacuating the ambient air from the system was imperative before light irradiation. A 300 W Xe lamp (light intensity: 280 mW/cm^−2^) with AM 1.5 filter was used as the light source. During light irradiation, continuous stirring of the reaction compounds was maintained using a magnetic stir bar. The system temperature was diligently maintained at 5 °C through the circulation of cooling water. The resulting hydrogen yield was quantified using an online gas chromatograph (GC7920, CeAulight, Beijing, China), and argon used as the carrier gas.

Furthermore, monochromatic light with specific wavelengths (400 nm, 420 nm, 450 nm, 500 nm, 550 nm, 600 nm, 650 nm, and 700 nm) was applied from a 300 W xenon lamp to quantify the apparent quantum efficiency (AQE). Average intensity per unit area was determined using a radiometer (CEL-NP2000-2A, CeAulight, Beijing, China), followed by AQE calculation through the subsequent formula:(2)$$AQE=\frac{2\times the\;number\;of\;hydrogen\;molecules\;produced}{the\;number\;of\;incident\;photon}\times 100\%$$

### 2.4. Photoelectrochemical Evaluation

The photoelectrochemical properties were examined using a conventional three-electrode system in combination with an electrochemical workstation (CHI 760e, Chenhua, Shanghai, China). The electrolyte was a 0.5 M Na_2_SO_4_ solution. The reference electrode was used as a saturated Ag/AgCl electrode, while the counter electrode was a platinum wire. To prepare the working electrode, a dispersion liquid was formed by employing 5 mg of the sample in a solvent mixture of 0.4 mL of water and 0.6 mL of isopropanol. Subsequently, 100 µL of this dispersion liquid was applied onto an ITO glass substrate and subjected to vacuum drying. A light source comprising an AM 1.5 filter using a xenon lamp with a power of 300 W was employed. For electrochemical impedance spectroscopy (EIS), an AC perturbation of 5 mV was applied, spanning a frequency from 10^−2^ to 10^5^ Hz. Mott–Schottky plots were generated at frequencies of 500 Hz and 1000 Hz, scan rate of 2 mV·s^−1^. Photocurrent response tests were executed under a constant potential of 0.1 V, with illumination by a xenon lamp with a power of 300 W (AM 1.5). The surface photovoltage (SPV) was assessed using the CEL-SPS1000 test system. This system comprises a lock-in amplifier (Stanford), chopper, monochromator (Stanford), and a 500 W xenon lamp as the light source.

### 2.5. Transient Absorption Experiments

A custom-built spectrometer was employed for conducting femtosecond transient absorption (TA) measurements. The fundamental pulses were obtained from a Ti: Sa amplified laser system (Legend Elite-1K-HE, Coherent, Santa Clara, CA, USA). The laser delivered 25 fs pulses at 1 kHz, and the output was split for white-light continuum generation and optical pumping in the UV region. The excitation wavelength was obtained from a tunable optical parametric amplifier (TOPAS-C, Light Conversion, Vilnius, Lithuania) and selected at 380 nm pump pulse for excitation. The generation of supercontinuum white light with 800 nm pumped CaF_2_ results in a spectral window spanning from 350 nm to 800 nm. Control over the delay in time between the pump and probe pulses was achieved through the application of an optical delay stage. A fiber spectrometer (AvaSpec-ULS2048CL-EVO, Avantes, Apeldoorn, The Netherlands) was used to collect visible transient absorption probe signals. Throughout the experiments, the intensity of the 380 nm laser pulses used for pumping was consistently maintained at 400 nJ. Then, the femtosecond TAS data were input into the “Glotaran” software (Glotaran 1.5.1) for global fitting [34,35]. Glotaran is a graphical user interface (GUI) based on the R-package TIMP for global and target analysis of time-resolved spectroscopy. Based on the singular value decomposition (SVD), the evolution-associated spectra were generated using a sequential kinetic model.

## 3. Results and Discussion

### 3.1. Morphology and Structural Characterization

A detailed depiction of the morphological characteristics of the C_60_-CuPcOC_8_ is presented in Figure 1. A rod-like structure with diameters ranging from 100 to 150 nm was revealed by SEM. This unique morphology stands in contrast to the individual C_60_ and CuPcOC_8_ (Appendix A). Furthermore, a uniform crystal lattice pattern of C_60_-CuPcOC_8_ was elucidated by HRTEM in Figure 1b. The interplanar spacing of C_60_-CuPcOC_8_ was 0.68 nm, 0.99 nm, and 1.47 nm, corresponding to the (301), (101), and (200) crystallographic planes, respectively [36,37,38]. Subsequently, the PXRD profiles confirm that the identified peaks are 13.4°, 8.96°, and 5.98° (Figure 1c), aligning with the HRTEM analysis.

The molecular frontier orbitals of the C_60_-CuPcOC_8_ are further computed using density functional theory (DFT) at the b3lyp/6-31g(d) level [39]. The HOMO and LUMO of C_60_-CuPcOC_8_ are situated in the phthalocyanine and the C_60_ segment, respectively (Appendix A). This D–A structure is helpful to promote charge separation [40]. A broad absorption spectrum spanning from 300 to 800 nm is exhibited by C_60_-CuPcOC_8_, as demonstrated in the UV-Vis DRS analysis (Figure 1d). In contrast to C_60_ and CuPcOC_8_, the absorption onset of C_60_-CuPcOC_8_ undergoes broadening and a red shift. This can be attributed to the strong electron donor–acceptor interactions within C_60_-CuPcOC_8_, as well as the π–π stacking effect that enhances electronic wavefunction conjugation [41].

### 3.2. Efficient Photocatalytic Activity of the D-A Structure

The molecular orbitals engage in linear interactions via π–π interactions, resulting in the emergence of semiconductor energy bands [42]. The electrochemical Mott–Schottky analysis of C_60_-CuPcOC_8_ reveals positive slopes (Appendix A), indicating the n-type semiconductor. The flat band potential of C_60_-CuPcOC_8_ is estimated to be the conduction band (CB). Importantly, the computed CB value for C_60_-CuPcOC_8_ is −0.88 eV. It meets the thermodynamic conditions for hydrogen generation (E(H^+^/H_2_) = −0.41 eV, pH = 7) compared to the normal hydrogen electrode (NHE) reference at pH 7.

Following the optimization of catalyst and co-catalyst dosages (Appendix A), a notably efficient hydrogen evolution efficiency of 8.32 mmol·g^−1^·h^−1^ was achieved by employing C_60_-CuPcOC_8_ across the entire spectrum (Figure 2a). It surpassed the individual CuPcOC_8_ or C_60_ by two orders of magnitude (Figure 2b). The solubility of CuPc was improved in organic solvents by modifying CuPc with octoctyloxy. C_60_-CuPcOC_8_ assembly was obtained through the interfacial assembly of CuPcOC_8_ and C_60_, leading to a broad spectral absorption range of 300–800 nm. In comparison to other reported phthalocyanine-based organic photocatalysts (as summarized in Table 1), C_60_-CuPcOC_8_ exhibited remarkable photocatalytic performance, making it a promising candidate for hydrogen generation applications. Furthermore, the apparent quantum efficiency (AQE) of C_60_-CuPcOC_8_ was evaluated at various wavelengths to evaluate its capability for absorbing visible light. The wavelength-dependent AQE values of C_60_-CuPcOC_8_ closely paralleled the outcomes obtained from DRS. The trends of the AQE values at different single wavelengths are identical to those of the DRS absorption intensities. The outcomes suggest that the efficiency of light conversion is highly dependent on the range of light response [43].

Furthermore, stability plays a pivotal role in determining the feasibility of photocatalyst recovery and recycling processes. After undergoing five consecutive cycles of photocatalytic evaluation, the photocatalytic effectiveness of C_60_-CuPcOC_8_ was maintained at levels surpassing 95%, with no discernible deactivation observed (Figure 2d). Additionally, the structural integrity was confirmed through characterizations of XRD, XPS, and TEM analyses, as presented in Appendix A. These results emphasize the stability and sustainable nature of C_60_-CuPcOC_8_ as a photocatalyst for hydrogen generation.

### 3.3. Construction of the D–A Structure Establishes a Robust IEF

The IEF stands as the primary driving force governing the separation and transport of photogenerated charges [55]. A systematic investigation was conducted to understand the precise factors that contribute to the exceptional photocatalytic performance. The electrostatic potential map for C_60_-CuPcOC_8_ is depicted in Figure 3a [56]. The positive potentials are observed at the core of the C_60_ sphere, while negative potentials are concentrated within the copper phthalocyanine ring. A propensity for charge transfer from CuPcOC_8_ to C_60_ is suggested by this distribution, facilitating charge separation upon photoexcitation. Given the strong donor–acceptor interaction, the calculated dipole moment of the C_60_-CuPcOC_8_ molecular complex is approximately 2.90 Debye. This result is helpful to the formation of a robust IEF. It facilitates the rapid separation of photogenerated charge carriers. The strength of the IEF is quantified using the model developed by Kanataeet et al., as illustrated in Appendix A [57]. Surprisingly, the IEF strength within C_60_-CuPcOC_8_ is 16.86 times higher than CuPcOC_8_. This enhancement significantly contributes to the facilitation of the separation and transport of photogenerated charges. Ultimately, it leads to a noteworthy enhancement in the photocatalytic activity for hydrogen generation in C_60_-CuPcOC_8_ compared to CuPcOC_8_.

### 3.4. The D-A Structure Facilitates Charge Separation and Transport

Femtosecond transient absorption spectroscopy (TAS) measurements were conducted to explore the complex kinetics involved in the separation of photogenerated charges. Following laser irradiation at 380 nm, the early femtosecond TAS of C_60_-CuPcOC_8_ reveals excited-state absorption occurring at approximately 540 nm, concurrent with stimulated emission signals observed at 610 nm. Such characteristics are ascribed to the singlet excited state of CuPcOC_8_ [58,59]. Following this excited state evolution, a novel set of excited state absorption (ESA) features emerge in close proximity to 560 nm, 710 nm, and 1030 nm in Figure 4b. The distinctive TA peaks within the visible spectrum bear a remarkable resemblance to the CuPcOC_8_ cation radicals [60,61]. Simultaneously, the ESA signal was detected at 1030 nm in the near-infrared (NIR) region with the emergence of C_60_ anion radicals [62]. This offers compelling evidence for the intrinsic occurrence of photo-induced charge separation phenomena within the C_60_-CuPcOC_8_ system. By employing a global analysis, through the utilization of a sequential kinetic model for a comprehensive analysis, three distinct kinetic components are elucidated in Figure 4c,d [63]. The initial phase is attributed to a “hot” exciton state stemming from the high-energy CuPcOC_8_ material within the instrumental response window of approximately ~0.3 picoseconds [64]. Subsequently, this “hot” exciton state expeditiously propels charge separation, occurring within 0.6 ps [65]. The charge-separated state C_60_^·−^-CuPcOC_8_^·+^ (CS1) state undergoes evolution into a third component within approximately 29 ps, identified as the long-lived charge-separated state C_60_^·−^-CuPcOC_8_^·+^ (CS2), with a determined lifetime of 0.94 ns. On the contrary, the femtosecond TAS of CuPcOC_8_ exhibits characteristics indicative of localized excited states, as illustrated in Appendix A, without charge separation. Therefore, according to the TAS study, ultra-fast and long-lived charge separation do exist in C_60_-CuPcOC_8_.

Besides charge separation, the charge transport property holds equal significance in the process of photocatalysis. The electrochemical impedance spectra were initially examined in Figure 4a. The charge transfer impedance (R_ct_) was indicated by the diameter of the arc in the EIS results [66]. Notably, a significantly reduced R_ct_ was observed for C_60_-CuPcOC_8_ compared to CuPcOC_8_, effectively enhancing its charge transport capability. Upon exciting the CuPcOC_8_ and C_60_-CuPcOC_8_ assemblies at 600 nm, a quenched fluorescence absorption signal was observed from CuPcOC_8_ at 710 nm (Figure 4b). This phenomenon was attributed to the promotion of charge separation by the D–A structure of the C_60_-CuPcOC_8_ assemblies. Consequently, the excited electrons no longer underwent radiative transitions but returned to the ground state through a charge transfer or energy transfer process instead. Additionally, the time–current curves of C_60_-CuPcOC_8_ and CuPcOC_8_ under steady potential conditions are depicted in Figure 4c. In comparison to CuPcOC_8_, a more robust photocurrent response is exhibited by C_60_-CuPcOC_8_. The enhanced response is attributed to the improvement of charge transfer efficiency. In the surface photovoltage (SPV) spectrum (Figure 4d), a significantly elevated positive photovoltage signal is observed in the 300 to 520 nm range for C_60_-CuPcOC_8_. Notably, this signal is approximately ten times stronger than that observed in CuPcOC_8_. This notable difference implies a substantial increase in the migration of photogenerated charges toward the surface within the C_60_-CuPcOC_8_ assembly.

Drawing upon the preceding discussion, a plausible mechanism for the photocatalytic is proposed in Figure 5. The strong D–A interactions of C_60_-CuPcOC_8_ play a pivotal role in establishing a robust IEF. The robust IEF facilitates charge separation and transfer. Photogenerated electrons migrate to the surface, accelerating a reduction reaction yielding hydrogen, wherein Pt functions as a co-catalyst. Concurrently, the photogenerated holes engage in reactions with the sacrificial agent, ascorbic acid. In brief, the C_60_-CuPcOC_8_ system achieves efficient separation of photogenerated electron–hole pairs, resulting in an efficient photocatalytic hydrogen generation.

## 4. Conclusions

In summary, we have utilized the light-harvesting CuPcOC_8_ donor and efficient-electron-transport C_60_ acceptor to synthesize a novel photocatalyst, C_60_-CuPcOC_8_. The photocatalyst exhibits a broad spectral absorption range spanning from 300 to 800 nm, facilitating excellent photocatalytic hydrogen generation. The incorporation of a D–A architecture within C_60_-CuPcOC_8_ results in a larger molecular dipole moment, facilitating the creation of a robust IEF. This IEF promoted the separation and transportation of the photogenerated charge carriers. Consequently, C_60_-CuPcOC_8_ demonstrates exceptional hydrogen evolution activity, achieving a rate of 8.32 mmol·g^−1^·h^−1^. This rate surpasses that of C_60_ and CuPcOC_8_ monomers by two orders of magnitude. This research introduces a novel approach for designing highly active supramolecular photocatalytic materials, offering valuable insights for the construction of an efficient photocatalytic hydrogen generation system.

## Data Availability

The data presented in this study are available upon reasonable request from the corresponding author.

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
