# Peer review of "Efficient Charge Separation and Transport in Fullerene-CuPcOC8 Donor–Acceptor Nanorod Enhancing Photocatalytic Hydrogen Generationâ€"

_nanomaterials, 2024, doi:10.3390/nano14030256_

Round 1
Reviewer 1 Report
Comments and Suggestions for Authors
Efficient Charge Separation and Transport in Fullerene-CuPcOC8 Donor-Acceptor Nanorod Enhancing Photocatalytic Hydrogen Generation
Zihui Hua 1,2, Bo Wu 1,2*, Yuhe Zhang 1, Chong Wang 1,2, Tianyang Dong 1, Yupeng Song 2,3, Ying Jiang 1,2 and Chunru Wang 1,2*
1 Beijing National Laboratory for Molecular Sciences, Key Laboratory of Molecular Nanostructure and Nanotechnology, Institute of Chemistry, Chinese Academy of Sciences, Beijing 100190, China; zhhuaa@iccas.ac.cn
2 University of Chinese Academy of Sciences, Beijing 100049, China
3 Key Laboratory of Photochemical Conversion and Optoelectronic Materials and CityU-CAS Joint Laboratory of Functional Materials and Devices, Technical Institute of Physics and Chemistry, Chinese Academy of Sciences, Beijing, 100190 China
* Correspondence: zkywubo@iccas.ac.cn (B.W.); crwang@iccas.ac.cn (C.W.)
The generation of hydrogen by photocatalytic decomposition of water is a promising route to produce large-scale, cost-effective renewable hydrogen energy generation and address the global energy crisis. However, the rational design of efficient photocatalysts, leading to optimized light trapping capability, efficient charge separation and transport, and long-term stability, is crucial for achieving high yields in hydrogen generation.
In this work, authors have engineered organic semiconductor photocatalysts made of fullerene – Cu phthalocyanine (C60-CuPcOC8) donor-acceptor done using liquid-liquid precipitation method, achieving both efficient hydrogen generation and high stability. They show that significant donor-acceptor interaction leading to a large internal electric field, facilitates the efficient electron transfer from CuPcOC8 to C60 occurring within 0.6 ps. The rate of photocatalytic hydrogen generation is two orders of magnitude higher than the individual C60 and CuPcOC8 and remains above 95% even after 10 hours. This study provides a critical insight into advancing the field of photocatalysis.
The paper is well written. Methodologies are results are clearly presented and discussed. The fundamental mechanisms at the origin of the efficient charge separation are clear described. Authors show a remarkable efficiency of H2 production by combining C60 and CuPcOC8, regarding the comparable systems. The paper can be considered for publication after authors provide answers to the following minor questions listed below:
Introduction:
· A brief remind of authors’ objectives regarding the state of the art is missing at the end of introduction.
· I suggest to highlight in the end the remarkable efficiency of the proposed assembly compared to the literature.
Experimental:
· Part 2.1:
a. “The self-assemblies of C60 and CuPcOC8 were achieved using an identical method.” Authors mention the possibility to produce assemblies and self-assemblies of these organic compounds. Can the authors give more information about those self-assemblies, the formation mechanism, their interest as photocatalysts… later in the results and discussion section?
b. Details about the hole sacrificial agent are missing as well as the deposition of Pt co-catalysts on the molecular assembly.
· Part 2.2:
a. About UV-Vis diffuse reflectance spectra, why using BaSO4 as a reflectance material and not ITO as used for photoelectrochemical studies?
b. Are the organic compounds immobilized on the surface in the same way for both surface? Does the nature of the substrate affects molecular assembly/self-assembly?
· Part 2.5: Few clarification and additional details are necessary in this sub-section.
a. Authors mention the use of an amplifier to produce 380 nm pulse radiation. I guess it is an optical parametric amplifier. It could be mentioned.
b. It is not clearly said if the 380 nm pulse is the pump pulse or the seed for the probe pulse.
c. Nothing is given about the probe pulse that is a white light continuum in TA experiments. Can authors give some details about how is generated (from the 800 nm or the 380 nm pulse?) and its spectral width.
d. An optical fiber is used to collect probe pulses, does is mean that authors have a probe pulse for measurement and another one to reference the continuum spectral intensity?
e. It is mentioned that an electric delay line is used to change pump-probe delay. We usually employed an optical delay line to cover fs to several ns delay, while electrical delay line limited to the ns resolution are used for longer delays. Can the authors clarify this point?
f. Is it possible to explain what is the sequential kinetic model used to generated spectrum fits?
g. Are TA measurements performed in solution or on the deposited layer? If it is solution, what can be expected in the crystalline phase compared to isolated single D-A assembly?
Results and discussion:
· Part 3.1:
a. The follow sentence can be removed as it is already explained in the experimental section: “The synthesis of C60-CuPcOC8 assembly was accomplished through the liquid-liquid interface deposition method.”
b. HRTEM and XRD suggest a nice crystalline structure of the assembly. How the conformation of the alkyle chains is affect or affects the crystallization? What is the role of these chains?
c. About π-π stacking, I guess it is between carbone cycles of C60 and CuPc at the frontier between both molecules?
· Part 3.2:
a. Can authors remind what is the thermodynamic conditions for hydrogen generation and why a CB value of 0.88 eV meets this condition? It would help reader that in the domain of H2 production.
b. The sentence “The outcomes suggest that the photocatalytic hydrogen generation reaction is driven by absorbed incident light” line 180, is not clear to me. As a photocatalytic reaction, it is expected that it is driven by absorbed incident light.
· Part 3.3:
a. The electrostatic map was computed for an isolated D-A system. How the crystalline phase affects the IEF and the dipole moment intensity?
Conclusions:
· Did the author explore or can anticipate the possible to suppress the Pt co-catalyst by modifying the molecular assembly in order to have active sites for the reduction reaction of H2O into H2?
Supplementary:
· The exact definition of hole sacrificial agent acronyms used in table S1 would be useful.
Comments on the Quality of English Language
· Line 174 : “In comparison” instead of “In compared”
· Line 241 : “do exist” instead of “is existed”
Author Response
Reviewer 1
Comments:
The generation of hydrogen by photocatalytic decomposition of water is a promising route to produce large-scale, cost-effective renewable hydrogen energy generation and address the global energy crisis. However, the rational design of efficient photocatalysts, leading to optimized light trapping capability, efficient charge separation and transport, and long-term stability, is crucial for achieving high yields in hydrogen generation.
In this work, authors have engineered organic semiconductor photocatalysts made of fullerene–Cu phthalocyanine (C60-CuPcOC8) donor-acceptor done using liquid-liquid precipitation method, achieving both efficient hydrogen generation and high stability. They show that significant donor-acceptor interaction leading to a large internal electric field, facilitates the efficient electron transfer from CuPcOC8 to C60 occurring within 0.6 ps. The rate of photocatalytic hydrogen generation is two orders of magnitude higher than the individual C60 and CuPcOC8 and remains above 95% even after 10 hours. This study provides a critical insight into advancing the field of photocatalysis.
The paper is well written. Methodologies are results are clearly presented and discussed. The fundamental mechanisms at the origin of the efficient charge separation are clear described. Authors show a remarkable efficiency of H2 production by combining C60 and CuPcOC8, regarding the comparable systems. The paper can be considered for publication after authors provide answers to the following minor questions listed below:
Introduction:
A brief remind of authors’ objectives regarding the state of the art is missing at the end of introduction.
Response: Thanks for the valuable suggestion. We have added our objectives regarding the state of the art at the end of introduction in the revised manuscript. It can then be expected that C60-CuPcOC8 photocatalytic hydrogen generation activity will be significantly enhanced.
I suggest to highlight in the end the remarkable efficiency of the proposed assembly compared to the literature.
Response: Thanks for the valuable suggestion. We have added efficiency of the proposed assembly compared to the literature in the revised manuscript. However, the current efficiency of most Pc-based photocatalysts for hydrogen generation is unsatisfactory. In this study, a novel donor-acceptor (D-A) type photocatalyst C60-CuPcOC8 was devised with the aim of enhancing photocatalytic hydrogen generation. C60-CuPcOC8 demonstrates exceptional hydrogen evolution activity, achieving a rate of 8.32 mmol·g-1·h-1.
Experimental:
Part 2.1:
- “The self-assemblies of C60 and CuPcOC8 were achieved using an identical method.” Authors mention the possibility to produce assemblies and self-assemblies of these organic compounds. Can the authors give more information about those self-assemblies, the formation mechanism, their interest as photocatalysts… later in the results and discussion section?
Response: Thanks for the valuable suggestion. We have already added some details about the self-assemblies of C60 and CuPcOC8 in the revised manuscript. Self-assemblies are prepared by stacking two identical molecules through π-π interactions. These self-assemblies were prepared for comparison with the donor-acceptor assembly of C60-CuPcOC8. The hydrogen generation activity of C60-CuPcOC8 is two orders of magnitude higher than that of self-assemblies of C60 and CuPcOC8
- Details about the hole sacrificial agent are missing as well as the deposition of Pt co-catalysts on the molecular assembly.
Response: Thanks for the valuable suggestion. We have already added more details about the hole sacrificial agent and the deposition of Pt co-catalysts on the molecular assembly in Part 2.3. The hole sacrificial agent can consume photogenerated vacancies. Photodeposition is a technique based on light-induced electrochemistry. The process of photodeposition of Pt depends mainly on the photoelectrochemical reaction induced by photocatalysts. The equation for the reductive photodeposition of Pt is as follows.
Part 2.2:
- About UV-Vis diffuse reflectance spectra, why using BaSO4 as a reflectance material and not ITO as used for photoelectrochemical studies?
Response: Thanks for the kind suggestion. UV-Vis diffuse reflectance spectra are used to test the light absorption capacity of powder samples with an integrating sphere attachment. It comprises a spherical cavity with an inner wall coated with a highly reflective substance. The highly reflective material needs to satisfy the requirement that it should have good reflectance (100%) in the wavelength range to be measured, with no characteristic absorption. Commonly employed reflective materials, such as MgO, BaSO4, MgSO4, etc., are characterized by a reflectivity close to 1 (typically around 0.98-0.99). MgO is mechanically less robust compared to BaSO4, with BaSO4 emerging as the preferred reflective material in current applications.
- Are the organic compounds immobilized on the surface in the same way for both surface? Does the nature of the substrate affects molecular assembly/self-assembly?
Response: Thanks for the valuable suggestion. The organic compounds are immobilized on the surface in the same way for both surfaces. And the nature of the substrate does not affect molecular assembly/self-assembly. The assembly/self-assembly powders were obtained by liquid-liquid interfacial deposition. To conduct SEM testing, the prepared assembly/self-assembly were dispersed in ethanol to form a suspension, and then added dropwise onto a silicon wafer. Similarly, TEM testing was performed by adding the suspension dropwise onto a copper microgrid. For XRD testing, assembly/self-assembly powders were uniformly spread in a sample well. The assembly/self-assembly were prepared prior to testing, and the sample processing methods have no effect on the properties of the assembly/self-assembly. In addition, we have added the details on the treatment of the test samples in Experiment 2.2 section.
Part 2.5: Few clarification and additional details are necessary in this sub-section.
- Authors mention the use of an amplifier to produce 380 nm pulse radiation. I guess it is an optical parametric amplifier. It could be mentioned.
Response: Thanks for the kind suggestion. We have added details about the optical parametric amplifier in the revised manuscript. We used the optical parametric amplifier TOPAS-prime for transient absorption experiments to generate the pump light.
- It is not clearly said if the 380 nm pulse is the pump pulse or the seed for the probe pulse.
Response: Thanks for the valuable suggestion. The 380 nm pulse is the pump pulse, not the seed for the probe pulse.
- Nothing is given about the probe pulse that is a white light continuum in TA experiments. Can authors give some details about how is generated (from the 800 nm or the 380 nm pulse?) and its spectral width.
Response: Thanks for the kind suggestion. The generation of supercontinuum white light with 800 nm pumped CaF2 results in a spectral window spanning from 350 nm to 800 nm. We have added some details in the revised manuscript.
- An optical fiber is used to collect probe pulses, does is mean that authors have a probe pulse for measurement and another one to reference the continuum spectral intensity?
Response: Thanks for the valuable suggestion. We have a probe pulse for measurement and another one to reference the continuum spectral intensity. We have added the details in the revised manuscript.
- It is mentioned that an electric delay line is used to change pump-probe delay. We usually employed an optical delay line to cover fs to several ns delay, while electrical delay line limited to the ns resolution are used for longer delays. Can the authors clarify this point?
Response: Thanks for the insightful question. Actually, an optical delay line is used to change pump-probe delay. We have corrected in the revised manuscript.
- Is it possible to explain what is the sequential kinetic model used to generated spectrum fits?
Response: Thanks for the valuable suggestion. Glotaran is a graphical user interface (GUI) based on the R-package TIMP for global and target analysis of time-resolved spectroscopy. Based on the singular value decom-position (SVD), the Evolution Associated Spectra were generated using a sequential kinetic model.
- Are TA measurements performed in solution or on the deposited layer? If it is solution, what can be expected in the crystalline phase compared to isolated single D-A assembly?
Response: Thanks for the valuable suggestion. TA measurements are performed in aqueous dispersion. The powder of the assembly was dispersed in water to acquire a dispersion for transient absorption experiments. This approach is in line with the conditions employed for the experimental assessment of hydrogen production activity.
Results and discussion:
Part 3.1:
- The follow sentence can be removed as it is already explained in the experimental section: “The synthesis of C60-CuPcOC8 assembly was accomplished through the liquid-liquid interface deposition method.”
Response: Thanks for the valuable suggestion. We have removed this sentence in the revised manuscript.
- HRTEM and XRD suggest a nice crystalline structure of the assembly. How the conformation of the alkyle chains is affect or affects the crystallization? What is the role of these chains?
Response: Thanks for the valuable suggestion. The addition of alkyl chains may decrease the crystallinity. However, the purpose of these chains is to enhance the solubility of CuPc in organic solvents, allowing for its interfacial assembly with C60.
- About π-π stacking, I guess it is between carbone cycles of C60 and CuPc at the frontier between both molecules?
Response: Thanks for the kind suggestion. The C60 and CuPcOC8 centers are π-π stacked in the following manner.
Part 3.2:
- Can authors remind what is the thermodynamic conditions for hydrogen generation and why a CB value of 0.88 eV meets this condition? It would help reader that in the domain of H2 production.
Response: Thanks for the valuable suggestion. The condition for hydrogen production is that the conduction band energy level of the photocatalyst material is more negative than that of the reduction of hydrogen ions to hydrogen (E(H+/H2) = -0.41 eV). The conduction band energy level of C60-CuPcOC8 is -0.88 eV, which satisfies this condition. The original description contained a problem and we have modified in the revised manuscript.
- The sentence “The outcomes suggest that the photocatalytic hydrogen generation reaction is driven by absorbed incident light” line 180, is not clear to me. As a photocatalytic reaction, it is expected that it is driven by absorbed incident light.
Response: Thanks for the valuable suggestion. The trends of the AQE values at different single wavelengths are identical to those of the DRS absorption intensities. The outcomes suggest that the efficiency of light conversion is highly dependent on the range of light response. We have already made some explanation in the revised manuscript.
Part 3.3:
- The electrostatic map was computed for an isolated D-A system. How the crystalline phase affects the IEF and the dipole moment intensity?
Response: Thanks for the valuable suggestion. The electrostatic potential is a potential created by the distribution of charge between atoms within a molecule. It is determined by the distribution of atomic charges and the structure of the molecule. The molecular electrostatic potential can be used to describe the interactions between molecules as well as the reactions of molecules to external substances. The molecular dipole moment is a vector quantity that results from the product of the distance between the positive and negative charge centers and the amount of charge carried by them. The direction of the dipole moment is specified as pointing from the positive charge center to the negative charge center. The dipole moment of the molecular units is positively correlated with the magnitude of the internal electric field. Therefore, the strength of the internal electric field can be reflected by calculating the dipole moment of an isolated D-A system.
Conclusions:
Did the author explore or can anticipate the possible to suppress the Pt co-catalyst by modifying the molecular assembly in order to have active sites for the reduction reaction of H2O into H2?
Response: Thanks for the valuable suggestion. The active site for the hydrogen reduction reaction is Pt. The electrons are transferred to the co-catalyst Pt, and then undergoes a reduction reaction to produce hydrogen. We conducted hydrogen production activity tests both with and without Pt, and found that the hydrogen production activity was 1.31 mmol g-1 h-1 in the absence of Pt, which is lower than the activity observed in the presence of Pt.
Supplementary:
The exact definition of hole sacrificial agent acronyms used in table S1 would be useful.
Response: Thanks for the kind suggestion. We have already added the exact definition of hole sacrificial agent acronyms in the revised manuscript.
Comments on the Quality of English Language
Line 174 : “In comparison” instead of “In compared”
Response: Thanks for the kind suggestion. We have already made replacements in the revised manuscript.
Line 241 : “do exist” instead of “is existed”
Response: Thanks for the kind suggestion. We have already made replacements in the revised manuscript.

Reviewer 2 Report
Comments and Suggestions for Authors
Manuscript ID: nanomaterials-2804845; Title: “Efficient Charge Separation and Transport in Fullerene-CuPcOC8 Donor-Acceptor Nanorod Enhancing Photocatalytic Hydrogen Generation”. In this paper, the authors showcased a non-covalent supramolecular photocatalyst C60-CuPcOC8 was obtained using the liquid-liquid interface precipitation method. It exhibits a broad spectral response range spanning from 300 to 800 nm, allowing efficient utilization of solar energy. The rate of photocatalytic hydrogen generation for C60-CuPcOC8 is 8.32 mmol·g-1·h-1, which is two orders of magnitude higher than the individual C60 and CuPcOC8. The remarkable increase in hydrogen generation activity can be attributed to the development of a robust internal electric field within the C60-CuPcOC8 assembly. It is 16.68 times higher than that of the pure CuPcOC8. Additionally, the C60-CuPcOC8 manifests a robust IEF owing to the D-A interaction. This robust IEF promotes the efficient separation and transport of photogenerated charges, resulting in significantly enhanced photocatalytic activity for hydrogen generation.
The topic is interesting for the referee. Considering the importance of the current hot topic of solar energy research, the reviewer suggests that this manuscript can be accepted after the following minor issues are addressed.
1) The author needs to include the structure of photocatalyst C60-CuPcOC8 (C60 and CuPcOC8) for better understanding.
2) I think that the introduction is not sufficient, and the introduction should be elaborated with state of art. Needs to include an introduction to are Advantages and Disadvantages. and the author needs to include the motivation and necessity of this research.
3) Page No.2; The author needs to include an appropriate working principal diagram in the introduction.
4) Page No.2; 2.1. C60-CuPcOC8 Fabrication: I think that the author needs to include a graphical image for the fabrication method.
5) All equations should be represented with equation numbers. (ex: page No:3, line 111).
6) In the result and discussion, the author needs to include a comparison of the present work with the previous work. And needs to explain the importance of the present work.
7) I think that Table S1 should be moved to the main paper from SI.
Comments on the Quality of English Language
Minor editing of English language required. check typo errors.
Author Response
Reviewer 2
Comments:
Manuscript ID: nanomaterials-2804845; Title: “Efficient Charge Separation and Transport in Fullerene-CuPcOC8 Donor-Acceptor Nanorod Enhancing Photocatalytic Hydrogen Generation”. In this paper, the authors showcased a non-covalent supramolecular photocatalyst C60-CuPcOC8 was obtained using the liquid-liquid interface precipitation method. It exhibits a broad spectral response range spanning from 300 to 800 nm, allowing efficient utilization of solar energy. The rate of photocatalytic hydrogen generation for C60-CuPcOC8 is 8.32 mmol·g-1·h-1, which is two orders of magnitude higher than the individual C60 and CuPcOC8. The remarkable increase in hydrogen generation activity can be attributed to the development of a robust internal electric field within the C60-CuPcOC8 assembly. It is 16.68 times higher than that of the pure CuPcOC8. Additionally, the C60-CuPcOC8 manifests a robust IEF owing to the D-A interaction. This robust IEF promotes the efficient separation and transport of photogenerated charges, resulting in significantly enhanced photocatalytic activity for hydrogen generation.
The topic is interesting for the referee. Considering the importance of the current hot topic of solar energy research, the reviewer suggests that this manuscript can be accepted after the following minor issues are addressed.
1) The author needs to include the structure of photocatalyst C60-CuPcOC8 (C60 and CuPcOC8) for better understanding.
Response: Thanks for the valuable suggestion. We have added the structure of photocatalyst C60-CuPcOC8 (C60 and CuPcOC8) in the Scheme 2 in the revised manuscript.
2) I think that the introduction is not sufficient, and the introduction should be elaborated with state of art. Needs to include an introduction to are Advantages and Disadvantages. and the author needs to include the motivation and necessity of this research.
Response: Thanks for the valuable suggestion. We have added some introduction in the revised manuscript. There are inherent defects in most inorganic semiconductor photocatalysts, such as wide forbidden band widths, rapid photogenerated carrier recombination, inefficient utilization of sunlight, and photo-corrosion during the prolonged light exposure. These restrict their further development and application.
Copper phthalocyanine (CuPc) is an excellent organic pigment with high chemical stability to light and temperature. Due to these exceptional properties, a series of CuPc photocatalysts has been employed in the field of photocatalytic hydrogen generation.
However, the current efficiency of most Pc-based photocatalysts for hydrogen generation is unsatisfactory. In this study, a novel donor-acceptor (D-A) type photocatalyst C60-CuPcOC8 was devised with the aim of enhancing photocatalytic hydrogen generation.
3) Page No.2; The author needs to include an appropriate working principal diagram in the introduction.
Response: Thanks for the valuable suggestion. We have made an appropriate working principal diagram (Scheme 1) in the revised manuscript.
4) Page No.2; 2.1. C60-CuPcOC8 Fabrication: I think that the author needs to include a graphical image for the fabrication method.
Response: Thanks for the kind suggestion. We have added the fabrication of C60-CuPcOC8 (Scheme 2) in the revised manuscript.
5) All equations should be represented with equation numbers. (ex: page No:3, line 111).
Response: Thanks for the kind suggestion. We have represented all equations with equation numbers in the revised manuscript.
6) In the result and discussion, the author needs to include a comparison of the present work with the previous work. And needs to explain the importance of the present work.
Response: Thanks for the valuable suggestion. We have made some explanation in the revised manuscript. The solubility of CuPcOC8 was improved in organic solvents by modifying CuPc with octoctyloxy. C60-CuPcOC8 assembly was obtained through interfacial assembly of CuPcOC8 and C60, leading to a broad spectral absorption range of 300-800 nm. In comparison to other reported phthalocyanine-based organic photocatalysts (as summarized in Table 1), C60-CuPcOC8 exhibited remarkable photocatalytic performance, making it a promising candidate for hydrogen generation applications.
7) I think that Table S1 should be moved to the main paper from SI.
Response: Thanks for the kind suggestion. We have moved Table S1 to the main paper from SI in the revised manuscript.
Comments on the Quality of English Language
8) Minor editing of English language required. check typo errors.
Response: Thanks for the valuable suggestion. We have corrected typo errors in the revised manuscript.
